

**High-molecular-weight esters in α-pinene ozonolysis secondary organic**
**aerosol: Structural characterization and mechanistic proposal for their**
**formation from highly oxygenated molecules**
Ariane Kahnt[1,a], Reinhilde Vermeylen[1,b], Yoshiteru Iinuma[2,c], Mohammad Safi Shalamzari[1,d],
Willy Maenhaut[3], and Magda Claeys[1]
[1] Department of Pharmaceutical Sciences, University of Antwerp (Campus Drie Eiken), BE-
2610, Antwerp, Belgium
[2] Leibniz-Institut für Troposphärenforschung (TROPOS), Permoserstr. 15, D-04318 Leipzig,
Germany
[3] Department of Chemistry, Ghent University, Krijgslaan 281, S12, BE-9000 Ghent, Belgium
[a] now at: Development Bioanalysis, Janssen R&D, Turnhoutseweg 30, BE-2340 Beerse,
Belgium
[b] now at: Soil Service of Belgium, W. de Croylaan 48, BE-Heverlee, Belgium
[c] now at: Okinawa Institute of Science and Technology Graduate University, 1919-1 Tancha,
Onna-son, Kunigami, Okinawa 904-0495 Japan
[d] now at: Dr. Reddy's Laboratories Ltd., Zernikedreef 12, 2333 CL Leiden, The Netherlands
Correspondence to: M. Claeys (magda.claeys@uantwerpen.be)
**Abstract**
Stable high-molecular-weight esters are present in α-pinene ozonolysis secondary organic aerosol
(SOA) with the two most abundant ones corresponding to a diaterpenylic ester of *cis*-pinic acid
with a molecular weight (MW) of 368 $C_{19}H_{28}O_7$) and a hydroxypinonyl ester of *cis*-pinic acid
with a MW of 358 ($C_{17}H_{26}O_8$). However, their molecular structures are not completely elucidated
and their relationship with highly oxygenated molecules (HOMs) in the gas phase is still unclear.
In this study, liquid chromatography in combination with positive ion electrospray ionization
mass spectrometry has been performed on high-molecular-weight esters present in α-pinene/$O_3$





SOA with and without derivatization into methyl esters. Unambiguous evidence could be
obtained for the molecular structure of the MW 368 ester in that it corresponds to an ester of *cis*-
pinic acid where the carboxyl substituent of the dimethylcyclobutane ring and not the
methylcarboxyl substituent is esterified with 7-hydroxypinonic acid. The same linkage was
already proposed in previous work for the MW 358 ester (Yasmeen et al., 2010), but could be
supported in the present study. Guided by the molecular structures of these stable esters, we
propose a formation mechanism from gas-phase HOMs that takes into account the formation of
an unstable $C_{19}H_{28}O_{11}$ product, which is detected as a major species in α-pinene ozonolysis
experiments as well as in the pristine forest atmosphere by chemical ionization – atmospheric
pressure ionization – time-of-flight mass spectrometry with nitrate clustering (Ehn et al., 2012,
2014). It is suggested that an acyl peroxy radical related to *cis*-pinic acid ($RO_2·$) and an alkoxy
radical related to 7- or 5-hydroxypinonic acid (R'O·) serve as key gas-phase radicals and
combine according to a $RO_2· + R'O· → RO_3R'$ radical termination reaction. Subsequently, the
unstable $C_{19}H_{28}O_{11}$ HOM species decompose through the loss of oxygen or ketene from the inner
part containing a labile trioxide function and the conversion of the unstable acyl hydroperoxide
groups to carboxyl groups, resulting in stable esters with a molecular composition of $C_{19}H_{28}O_7$
(MW 368) and $C_{17}H_{26}O_8$ (MW 358), respectively. The proposed mechanism is supported by
several observations reported in the literature. On the basis of the indirect evidence presented in
this study, we hypothesize that $RO_2· + R'O· → RO_3R'$ chemistry is at the underlying molecular
basis of high-molecular-weight ester formation upon α-pinene ozonolysis and may thus be of
importance for new particle formation and growth in pristine forested environments.





## 1 Introduction

The molecular characterization of secondary organic aerosol (SOA) has been a topic of interest in atmospheric chemistry for the last decades, owing to the importance of organic aerosol in air quality and climate (for a review, see Nozière et al., 2015). SOA comprises a large number of oxygenated organic compounds, is a major constituent of submicrometer atmospheric particulate matter (PM), and both biogenic (e.g., isoprene, monoterpenes, sesquiterpenes) and anthropogenic (aromatics, *n*-alkanes) volatile organic compounds (VOCs) serve as precursors for SOA. Abundant biogenic VOCs in the terrestrial atmosphere are monoterpenes, having an annual global emission rate of 155 Tg with α-pinene as the major terpene emitted (Guenther et al., 2012). Several multifunctional SOA compounds, including monomers and dimers from α-pinene oxidation have been structurally identified (for a review, see Nozière et al., 2015). Recently, "extremely low-volatility organic compounds" (ELVOCs), currently termed "highly oxygenated molecules" (HOMs), originating from α-pinene ozonolysis have been detected in both laboratory and field experiments by chemical ionization – atmospheric pressure ionization – time-of-flight (CI-APi-TOF) mass spectrometry with nitrate clustering (Ehn et al., 2012, 2014; Zhao et al., 2013) and have received much attention because of their role in driving new particle formation and growth in pristine forested environments. Molecular characterization of α-pinene SOA constituents is needed to elucidate the underlying formation mechanism and establish its link with gas-phase HOMs, and efforts in this direction have recently been undertaken (Mutzel et al., 2015; Zhang et al., 2015; Krapf et al., 2016; Zhang et al., 2017). However, the relationship of HOMs detected in the gas phase upon α-pinene ozonolysis with stable high-molecular-weight SOA constituents is unclear, so that there is still a missing element in closing the α-pinene SOA system.

High-molecular-weight esters have been reported in α-pinene/O$_3$ SOA but their detailed chemical structures are only partially elucidated and their mechanism of formation is still elusive. A high-molecular-weight compound with a molecular weight (MW) of 358 has been reported for the first time by Hofmann et al. (1998) in α-pinene/O$_3$ SOA using off- and online mass spectrometry (MS). With online atmospheric pressure chemical ionization (APCI) MS it was shown that this compound is formed concomitantly with two monomers, i.e., *cis*-pinic acid and a MW 172 compound that was tentatively identified as norpinic acid. Tandem MS on the deprotonated





compound (*m/z* 357) revealed that it has a *cis*-pinic acid residue (*m/z* 185) as well as a *m/z* 171
residue. Later work by Müller et al. (2008) focused on the structure of the MW 368 compound. It
was shown that this compound is composed of *cis*-pinic and hydroxypinonic acid parts, which are
linked together by an ester bridge. The structure of the MW 358 compound was also addressed by
Yasmeen et al. (2010), who revised the structure of this compound and presented evidence that it
is a diaterpenylic ester of *cis*-pinic acid. The same conclusion was reached by Gao et al. (2010),
who also showed that the MW 358 ester is a major product in α-pinene ozonolysis experiments
performed at low mass loadings. Recent work by Beck and Hoffmann (2016), where use was
made of derivatization to the *n*-butylesters and subsequent tandem MS analysis of the lithiated
and ammoniated molecules, supported the structure of the MW 358 ester as a diaterpenylic ester
of *cis*-pinic acid. Furthermore, the MW 358 ester was detected as a major tracer in β-pinene
ozonolysis SOA characterization studies (Iinuma et al., 2007; Yasmeen et al., 2010).
It is noted that prior to the studies by Müller et al. (2008) and Yasmeen et al. (2010) several
studies dealt with the molecular characterization of high-molecular-weight compounds and that
very different possible structures have been advanced. Gao et al. (2004) assigned the MW 358 α-
pinene/$O_3$ compound to a dehydration product formed between the gem-diol forms of two
norpinonic acid molecules. Iinuma et al. (2004) reported MW 354 and 370 α-pinene/$O_3$ products
that were enhanced in acidic conditions and tentatively assigned them to reaction products
between the gem-diol of pinonaldehyde and pinonaldehyde, and between pinonaldehyde and
hydroxypinonaldehyde, respectively. Docherty et al. (2005) proposed peroxycarboxylic acid
dimers for the structure of higher-MW SOA products from the ozonolysis of α-pinene in which
peroxypinic acid and the gem-diol of a keto or aldehydic compound are connected via a peroxy
bridge. Tolocka et al. (2004) characterized high-molecular-weight compounds in α-pinene
ozonolysis SOA and suggested that the products were most likely formed by aldol and/or gem
diol formation. In addition, Witkowski and Gierczak (2014) explained the formation of MW 338
and 352 compounds in α-pinene ozonolysis as aldol reaction products of α-acyloxyhydroperoxy
aldehydes. All the above mentioned studies thus provide evidence that the structure elucidation of
high-molecular-weight α-pinene/$O_3$ compounds has turned out to be very challenging.
With regard to the structure elucidation of the MW 358 ester there is still ambiguity, in that two
positional isomers are possible (Fig. 1), and that different positional isomers have been proposed





by Yasmeen et al. (2010) [structure (**a**)], Gao et al. (2010) [structure (**b**)], and Beck and
Hoffmann (2016) [structure (**b**)]. Based on the MS data obtained it is not possible to
unambiguously support the structure of one or the other positional isomer. This issue will be
further addressed in Section 3. The same ambiguity holds for the MW 368 ester (Fig. 1). In
addition to the MW 358 and 368 esters, minor high-molecular-weight compounds (i.e., MWs
272, 300, 308, 312, 314, 326, 338, 344, 352, 356, 376, 378 and 400) have also been reported in α-
pinene/$O_3$ SOA (Müller et al., 2008; Yasmeen et al., 2010; Kourtchev et al., 2014; Witkowski
and Gierczak, 2014; Zhang et al., 2015) but these products will not be addressed in the present
paper.
High-molecular-weight esters have been detected up till now in many field studies that were
conducted in forested regions. MW 358 and 368 esters were first reported in ambient nighttime
PM with an aerodynamic diameter $\leq$ 2.5 µM ($PM_{2.5}$) that was collected at K-puszta, Hungary,
during a 2006 summer campaign (Yasmeen et al., 2010). They were later detected in several field
studies that were conducted in other forested environments (Kristensen et al., 2013, 2016;
Kourtchev et al., 2014, 2015). It was shown by Kourtchev et al. (2016) that oligomers (i.e.,
hetero-oligomers) are of climatic relevance in that elevated SOA mass is one of the key drivers of
oligomer formation not only in laboratory experiments but also in the ambient atmosphere. It was
also demonstrated in the latter study that the oligomer content is strongly correlated with cloud
condensation nuclei activities of SOA particles. Furthermore, it could be demonstrated in
laboratory chamber experiments that the ratio monomers/oligomers and the oligomer content in
α-pinene ozonolysis SOA are enhanced at low temperature and low precursor concentrations,
conditions that are relevant for the upper troposphere (Huang et al., 2017).
Efforts to understand ester formation from α-pinene ozonolysis have also been actively
undertaken. Yasmeen et al. (2010) proposed that ester formation took place in the particle phase
by esterification of *cis*-pinic acid with terpenylic acid but this mechanism was not retained in
later studies. Kristensen et al. (2014) demonstrated their formation through gas-phase ozonolysis
and supported the participation of a stabilized Criegee intermediate, as previously suggested for
the formation of unstable high-molecular-weight compounds that play a role in new particle
formation (Ziemann, 2002; Bonn et al., 2002; Lee and Kamens, 2005). In a study by Zhang et al.
(2015), the dynamics of particle-phase components of α-pinene SOA formation were investigated





in detail. It was shown that formation of monomeric products like *cis*-pinic acid is observed after
the consumption of α-pinene upon ozonolysis, which cannot be explained solely by a gas-phase
mechanism and points to a particle-phase mechanism. A mechanism involving gas-phase radical
combination of acyl peroxy radicals and a condensed-phase rearrangement was proposed that
potentially explains the α-pinene SOA features in terms of molecular structure, abundance,
growth rates, evolution patterns, and responses to variations in temperature, relative humidity,
and oxidant type. Furthermore, a recent study by Zhang et al. (2017), using ozonolysis of
deuterium-labeled α-pinene, demonstrated that hydroperoxy derivatives of pinonic acid
containing the hydroperoxy group at different positions are components of HOMs that are present
in the particle phase. In work prior to the above cited investigations other studies already
suggested the involvement of acyl peroxy radicals in the formation of HOMs upon α-pinene
ozonolysis (Ziemann, 2002; Docherty et al., 2005). In addition, evidence for peroxyhemiacetal
formation upon α-pinene ozonolysis has also been reported (Hall and Johnston, 2012a). All the
above cited studies thus document that establishing the underlying molecular mechanism leading
to ester formation in α-pinene ozonolysis is challenging. This is mainly due to a lack of
knowledge (or only a partial knowledge, i.e., molecular formulae) of the molecular structures of
both gas-phase intermediates and particulate-phase end products.
In the present paper, we focus on the structural characterization of the MW 358 and 368 esters
that are present in α-pinene/O$_3$ SOA. To this aim, we have performed liquid chromatography/
electrospray ionization mass spectrometry (LC/ESI-MS) in the positive ion mode on α-pinene/O$_3$
SOA with and without derivatization into methyl esters. A soft methylation procedure using
ethereal diazomethane was selected to avoid hydrolysis of the ester function present in the
targeted hetero-dimers. The aim of the methylation was two-fold: on the one hand, to confirm the
number of free carboxyl functions, and on the other hand, to obtain mass spectrometric
fragmentation that is different from that of intact esters in (+)ESI and to that obtained in previous
studies on intact esters in (–)ESI (Müller et al., 2008; Yasmeen et al., 2010; Zhang et al., 2015).
Led by the molecular structures of the MW 368 and MW 358 esters, we propose a formation
mechanism that takes into account the detection of a C$_{19}$ HOM in the gas phase by CI-APi-TOF
MS with nitrate clustering (Ehn et al., 2012, 2014; Zhao et al., 2013) and involves the
combination of an acyl peroxy radical related to *cis*-pinic acid with an alkoxy radical related to
isomeric hydroxypinonic acids, which are, as *cis*-pinic acid, major monomers in α-pinene SOA.



## 2. Experimental

### 2.1 α-pinene/O₃ chamber aerosol

Filter samples from α-pinene ozonolysis were obtained from experiments carried out in the 19 m³
TROPOS aerosol chamber at 50% relative humidity and 21 °C. 1.6 ppm α-pinene was reacted
with 615 ppb ozone without seed particles and no OH radical scavenger was added. The aerosol
formed was sampled after about one hour of reaction time using a quartz fibre filter, and the
sample was stored at –22 °C before analysis.

### 2.2 Filter sample preparation for analysis

A quarter of the chamber aerosol filter was extracted using three times 10 mL methanol and
applying ultrasonic agitation for 3 min. The combined extracts were concentrated to about 1 mL
at 35 °C with a rotary evaporator, were transferred to a 1 mL reaction glass vial, and were blown
to dryness under a stream of nitrogen. The dried residue was reconstituted in 250 μL
methanol/water (50/50, v/v) and analyzed by LC/(+)ESI-MS to characterize the non-derivatized
dimers. Another quarter of the filter was similarly extracted but was further subjected to a
methylation procedure using ethereal diazomethane to derivatize free carboxylic acids into their
corresponding methyl esters. Diazomethane was freshly prepared using the precursor diazald
(99%, Sigma-Aldrich) according to a standard procedure (Furniss et al., 1989). 500 μL from the
ethereal diazomethane solution was added to the dried filter extract. After a reaction time of
about 5 min, the sample was dried under a gentle stream of nitrogen and reconstituted in 250 μL
methanol/water (50/50, v/v) for LC/(+)ESI-MS analysis of methylated compounds.

### 2.3 Chemical analysis

LC/ESI-MS analysis was carried out using a Surveyor Plus system (pump and autosampler)
(Thermo Scientific, San Jose, CA, USA) and the chromatographic separation for both the non-
and the derivatized filter extracts was performed on an Atlantis T3 column (2.1 x 150 nm, 3 μm
particle size, Waters, Milford, MA, USA). An injection volume of 10 μL was used and a flow
rate of 0.2 mL min$^{-1}$ was applied. The mobile phases consisted of (A) 50 mM ammonium
formate buffer with pH 3 and (B) methanol. A 65-min gradient was applied using the following
program: (B) was kept at 3% for 5 min, increased to 95% in 15 min and kept for 25 min,





followed by the reconditioning to 3% in 10 min and keeping (B) at 3% for 10 min. A linear ion
trap mass spectrometer (LXQ, Thermo Scientific, San Jose, CA, USA) operated in the positive
mode was used as the mass analyzer and details regarding operational and optimization
procedures can be found in Kahnt et al. (2014).

## 3. Results and Discussion

### 3.1. Characterization of the MW 358 and 368 high-molecular weight esters

### 3.1.1. Previous studies on $[M - H]^-$, $[M + NH_4]^+$ and $[M + Li]^+$ molecular species

For clarity, we summarize here selected MS data already reported in a previous study (Yasmeen
et al., 2010) that led to the structural characterization of the MW 358 ester from α-pinene
ozonolysis SOA as a diaterpenylic ester of *cis*-pinic acid. The data are given in Section 1 of the
supplement (Fig. S1 and Scheme S1). Only one MW 358 isomer was detected in α-pinene/$O_3$
SOA; upon $MS^2$ its deprotonated molecule $[M -H]^-$ fragments to product ions at *m/z* 185 and
171, which are attributed to *cis*-pinic and diaterpenylic acid, respectively. However, based on this
information alone the ester linkage cannot be firmly established since two positional isomers are
possible (Fig. 1). A minor MW 358 isomer was also detected in β-pinene ozonolysis SOA, which
was very similar to that observed from α-pinene but showed an additional $MS^2$ $[M – H]^-$ product
ion at *m/z* 189, which could best be explained with a positional isomeric structure [structure (**b**);
Fig 1]. $MS^2$ data for the latter product are presented in Fig. S2 and Scheme S2 of the supplement.
More recent work by Beck and Hoffmann (2016) involving fragmentation of lithiated and
ammoniated molecular species of the *n*-butylated derivative supported the structure of the MW
ester from α-pinene/$O_3$ SOA as a diaterpenylic ester of *cis*-pinic acid; however, these authors
suggested a positional isomeric structure [structure (**b**); Fig. 1] which is different from that
proposed by Yasmeen et al. (2010) [structure (**a**); Fig. 1]. The MS data obtained for the $[M +$
$NH_4]^+$ and $[M + Li]^+$ molecular species of the *n*-butylated derivative also do not enable
unambiguous differentiation between positional isomeric structures.
For both the MW 358 and 368 esters accurate mass measurements to obtain the molecular
compositions have also been performed in previous studies using (–)ESI (e.g., Zhang et al.,
2015), and are not repeated in the present study. The molecular compositions of the MW 358 and
368 esters are $C_{17}H_{26}O_8$ and $C_{19}H_{28}O_7$, respectively.



[Fig. 1]

### 3.1.2. Mass spectrometric behavior of the ammoniated underivatized MW 358 ester

LC chromatographic data obtained for underivatized α-pinene/$O_3$ SOA are provided in Fig. S3 of
the supplement. It can be seen that the MW 358 product signal in both the negative (*m/z* 357) and
positive ion mode (*m/z* 376 ) has about half the intensity of the *m/z* 367 (MW 368) signal, and
shows intensities in the same range as the monomers detected at *m/z* 171 (MW 172; terpenylic
acid), *m/z* 185 (MW 186; *cis*-pinic acid), and *m/z* 199 (MW 200; hydroxypinonic acids).
Selected LC/(+)ESI-MS data for the non-derivatized MW 358 ester with its proposed structure in
α-pinene/$O_3$ SOA are presented in Fig. 2 and Scheme 1. Fragmentation of the $[M + NH_4]^+$ leads
to the loss of ammonia (*m/z* 359), yielding $[M + H]^+$, and further loss of a molecule of water (*m/z*
341), which can occur at different positions. Abundant product ions are observed at *m/z* 173 and
187, which can be rationalized by processes located in the internal ester linkage. The *m/z* 169
product ion can be explained through protonation of the ester group (pathway **a**) or through a
hydrogen rearrangement (pathway **b**) resulting in protonated *cis*-pinic acid (*m/z* 187) and
subsequent loss of a molecule of water. However, it is noted that with a positional isomeric
structure product ions at the same *m/z* values could be expected. The *m/z* 173 ion results from a
hydrogen rearrangement in the inner ester part (pathway **c**), which can lose one or two molecules
of water, giving rise to *m/z* 155 and 137, respectively. It can also be seen that *m/z* 155 can lead to
the loss of CO giving rise to *m/z* 127. The *m/z* 169 ion fragments further through the loss of
water, leading to *m/z* 151; here, we expect that the loss of water proceeds more readily from
structure (**a**) (Fig. 1) as water elimination from structure (**b**) would lead to strain in the
dimethylcyclobutane ring. We therefore retain structure (**a**) as the most likely structure for the
major MW 358 ester present in α-pinene/$O_3$ SOA.
[Fig. 2]
[Scheme 1]

### 3.1.3. Mass spectrometric behavior of the ammoniated MW 358 ester trimethylated

**derivative**



LC chromatographic data obtained for methylated α-pinene/O₃ SOA are provided in Fig. S4 of
the supplement. It can be seen that the signal corresponding to the MW 358 ester detected at $m/z$
has a comparable intensity as that corresponding to the MW 368 ester detected at $m/z$ 414.
The mass shifts observed due to derivatization into methyl esters support that the MW 358
compound contains three carboxyl groups, while the MW 368 compound contains two such
groups. The corresponding methylated monomers, i.e., terpenylic acid (detected at $m/z$ 204), *cis*-
pinic acid (detected at $m/z$ 232) and hydroxypinonic acid (detected at $m/z$ 232) show intensities in
the same range as the methylated MW 358 and 368 esters.
Selected LC/(+)ESI-MS data for the derivatized MW 358 ester with its proposed structure in α-
pinene/O₃ SOA are presented in Fig. 3 and Scheme 2. Fragmentation of the $[M + NH_4]^+$ ion ($m/z$
418) leads to the formation of three product ions at $m/z$ 201, 169 and 141, while further
fragmentation of $m/z$ 201 upon MS³ mainly leads to $m/z$ 169, and MS⁴ of the generated $m/z$ 169
mainly results in $m/z$ 141. Two different structures can be proposed for $m/z$ 201; structure (**a**) can
be explained following the loss of ammonia and ionization (protonation) at the inner ester
linkage, while structure (**b**) can be rationalized by a hydrogen rearrangement in the inner ester
linkage and loss of ammonia. Further loss of methanol (32 u) from $m/z$ 201 results in $m/z$ 169,
with two possible structures (**c**) and (**d**). It can be seen that structures (**c**) and (**d**) can give rise to
the loss of CO, resulting in $m/z$ 141. The weak ion at $m/z$ 137 can be explained by fragmentation
of $m/z$ 169 [structure (**c**)] through loss of methanol. It is noted that the same product ions could be
explained from the positional isomeric structure of the derivatized MW 358 ester; however, in
this case we would expect a more abundant $m/z$ 151 product ion, due to a more favorable loss of
water in the carboxymethyl terminus. Loss of a molecule of water from $m/z$ 169 [structure (**d**)]
leads to a weak product ion at $m/z$ 151, while further loss of a molecule of ketene also results in
$m/z$ 109.
[Fig. 3]
[Scheme 2]

### 280    3.1.4. Mass spectrometric behavior of the ammoniated underivatized MW 368 ester

Selected LC/(+)ESI-MS data for the ammoniated non-derivatized MW 368 ester with its
proposed structure in α-pinene/O₃ SOA are presented in Fig. 4 and Scheme 3. Fragmentation of





the $[M + NH_4]^+$ upon $MS^2$ leads to the loss of ammonia ($m/z$ 369), yielding $[M + H]^+$, and
product ions at $m/z$ 351, 333, 307, 183 and 169, of which $m/z$ 351 is the base peak, and essentially
the same pattern is seen upon $MS^3$ of $m/z$ 369. The product ions at $m/z$ 351 and 333 in the higher
mass range can simply be explained by the loss of one and two molecules of water, respectively.
The loss of $CO_2$ (44 u) leading to $m/z$ 307 is difficult to explain from a carboxy terminus and
likely takes place from the inner ester linkage. The product ion at $m/z$ 169 can be rationalized
through protonation at the inner ester function (route **a**) and further fragments through loss of
water ($m/z$ 151), as can be seen upon $MS^3$. Similarly, the product ion at $m/z$ 183 can arise through
protonation at the inner ester function (route **b**) and further loss of water results in $m/z$ 165. A
positional isomeric structure (due to loss of water from the left terminus) can also be suggested
for $m/z$ 351. The ions at $m/z$ 169 and 183 can also occur after loss of water from the left and right
carboxyl terminus, respectively. It is noted that all the ions discussed above can also be explained
with a positional isomeric structure [Fig. 1; structure (**a**)], although we would expect that such a
structure would lead to a less pronounced loss of water from $m/z$ 369 resulting in $m/z$ 351, as it
would result in strain in the dimethylcyclobutane ring.
[Fig. 4]
[Scheme 3]
**3.1.5. Mass spectrometric behavior of the ammoniated MW 368 ester dimethylated**
**derivative**
Selected LC/(+)ESI-MS data for the ammoniated MW 368 ester dimethyl derivative with its
proposed structure in α-pinene/$O_3$ SOA are presented in Fig. 5 and Scheme 4. Fragmentation of
the $[M + NH_4]^+$ ($m/z$ 414) upon $MS^2$ leads to the loss of ammonia ($m/z$ 397), yielding $[M + H]^+$,
and product ions at $m/z$ 379, 365, 269, 251, 201, 197, 183, 179, 165, 139 and 119, and essentially
the same pattern is seen upon $MS^3$ of $m/z$ 397. The product ions at $m/z$ 379 and 365 in the higher
mass range can simply be explained by the loss of a molecule of water and methanol,
respectively, of which the loss of water is due to an enolized keto group and that of methanol can
occur at one of the two methyl ester termini. The product ions at $m/z$ 201 and 183, observed upon
$MS^2$ of $m/z$ 414 and $MS^3$ of $m/z$ 397, can be explained through ionization at the inner ester
linkage and a hydrogen rearrangement. It is noted that these two product ions do not allow



differentiating between positional isomers of the MW 368 ester dimethyl derivative. The product
ions at *m/z* 269 and 251, observed upon MS$^2$ of *m/z* 414 and MS$^3$ of *m/z* 397, can be explained by
a cross-ring cleavage in the dimethylbutane ring, a fragmentation that has been observed in
previous studies for deprotonated *cis*-pinic acid (Yasmeen et al., 2011) and deprotonated *cis*-
norpinic acid (Yasmeen et al., 2010), both containing a keto group in α-position to the
dimethylcyclobutane ring. This fragmentation can be regarded as characteristic for one of the
positional isomeric forms, namely, structure (**b**) (Fig. 1), as it cannot be explained with the other
positional isomeric form (**a**). Further fragmentation upon MS$^3$ of *m/z* 379 leads to *m/z* 251, 179
and 119, which can be rationalized by the loss of propenoic acid (72 u), and the subsequent
combined loss of methanol and carbon monoxide. Thus, the MS data obtained for the
ammoniated MW 368 ester dimethylated derivative unambiguously support structure (**b**).
[Fig. 5]
[Scheme 4]
**3.2. Possible formation mechanism for the MW 368 and MW 358 esters**
**3.2.1. General mechanistic considerations**
It is noted that formation mechanisms involving unstable intermediates are generally hard to
formulate as unstable compounds cannot be isolated and structurally characterized; however, the
molecular structure of the gas-phase precursor (in this case, α-pinene), its known gas-phase
chemistry, the molecular composition of unstable intermediates and the molecular structure of
stable end products observed in the particle phase can provide crucial insights. Guided by the
molecular structures of the MW 368 [Fig. 1; structure (**b**)] and MW 358 esters [Fig. 1; structure
(**a**)] a formation mechanism is suggested, thereby taking into account that a C$_{19}$ HOM has been
detected as a major high-molecular-weight species in the gas phase upon α-pinene ozonolysis by
CI-APi-TOF MS with nitrate clustering (Ehn et al., 2012, 2014; Zhao et al., 2013; Krapf et al.,
2016). In an effort to propose pathways that lead to the formation of the MW 368 and 358 esters,
we have tried to formulate a uniform mechanism in that it involves the same acyl peroxy radical
related to *cis*-pinic acid and an alkoxy radical related to isomeric hydroxypinonic acids.
**3.2.2. Formation mechanism proposed for the MW 368 ester**



A possible formation mechanism leading to the MW 368 ester is outlined in Scheme 5. It is
suggested that an alkoxy radical related to 7-hydroxipinonic acid (**a**) (*cis*-2,2-dimethyl-3-
hydroxyacetylcyclobutylethanoic acid; for labeling, see Scheme S3 of the supplement) (R'O·)
and an acyl peroxy radical related to *cis*-pinic acid (**b**) (RO$_2$·) serve as key intermediates. Radical
termination according to a RO$_2$· + R'O· → RO$_3$R' reaction leads to a  HOM with a molecular
composition of C$_{19}$H$_{28}$O$_{11}$ (**c**), which corresponds to a major gas-phase species upon α-pinene
ozonolysis (Ehn et al., 2012, 2014; Krapf et al., 2016). The proposed mechanism is inspired by
the mechanism suggested by Zhang et al. (2015) to explain the formation of a MW 326 ester,
where two peroxy radicals related to *cis*-pinic acid combine according to a RO$_2$· + RO$_2$·→ ROOR
+ O$_2$ reaction. Further degradation of C$_{19}$H$_{28}$O$_{11}$ (**c**) involving the labile inner part containing a
linear trioxide bridge through the loss of oxygen and conversion of the acyl hydroperoxide
groups to carboxyl groups results in the MW 368 ester [Fig. 1; structure (**b**)] with a molecular
composition of C$_{19}$H$_{28}$O$_7$ (**d**). It is noted that the formation of C$_{19}$H$_{28}$O$_7$ corresponds to a RO$_2$· +
R'O· → ROR' + O$_2$ reaction, bearing similarity with the RO$_2$· + RO$_2$·→ ROOR + O$_2$ reaction
where the R groups are alkyl peroxy groups, which is known to involve a tetroxide intermediate
(e.g., Bohr et al., 1999). As to the formation of a linear trioxide intermediate (**c**), trioxides
containing a –(C=O)OOO– function have been reported in the literature, e.g. tertiary alkyl peroxy
hydrogen phthalates have been synthesized and are used as catalysts for the polymerization of
vinyl compounds (Komai, 1971). Unstable intermediates formed from species (**c**) can also be
considered, owing to the conversion of one acyl hydroperoxy group (C$_{19}$H$_{28}$O$_{10}$), the conversion
of two acyl hydroperoxy groups (C$_{19}$H$_{28}$O$_9$), the loss of oxygen (C$_{19}$H$_{28}$O$_9$), and the loss of
oxygen combined with the conversion of one acyl hydroperoxy group (C$_{19}$H$_{28}$O$_8$). In this context,
such species have been detected in the gas phase by CI-APi-TOF MS with nitrate clustering in an
α-pinene ozonolysis flow tube experiment by Krapf et al. (2016). The alternative mechanism
leading to C$_{19}$H$_{28}$O$_{11}$ (**c**) involving an acyloxy radical related to *cis*-pinic acid and an alkyl peroxy
radical related to 7-hydroxipinonic acid is also theoretically possible but is not likely because of
the mesomeric stabilization in the acyloxy radical.
[Scheme 5]
With regard to the suggestion that an alkoxy radical related to 7-hydroxipinonic acid is a key
gas-phase radical, it should be noted that hydroxipinonic acids are major monomers in



α-pinene/O$_3$ SOA (Fig. S3). The detailed mechanism leading to the peroxy radical related to *cis*-
pinic acid (RO$_2$·) and the alkoxy radical related to 7-hydroxypinonic acid (R'O·) are given in
Scheme S4 of the supplement. The proposed RO$_2$· + R'O· → RO$_3$R' radical termination reaction
leads to a MW 368 ester with structure (**b**) (Fig. 1) [species (**d**) in Scheme 5], namely, an ester of
*cis*-pinic acid where the carboxyl substituent of the dimethylcyclobutane ring and not the
carboxymethyl group is esterified with 7-hydroxypinonic acid. It can also be seen that the labile
gas-phase intermediate (**c**) contains *cis*-pinic acid and 7-hydroxypinonic acid residues and thus
can serve as a precursor for these monomers and their corresponding hydroperoxides. In this
context, a recent study by Zhang et al. (2017) provided evidence for the formation of
7-hydroperoxypinonic acid from degradation of an unstable dimer precursor in α-pinene/O$_3$ SOA.
It is also worth mentioning that both *cis*-pinic acid (e.g., Yu et al., 1999; Glasius et al., 2000;
Larsen et al., 2001; Winterhalter et al., 2003) and 7-hydroxypinonic acid (e.g., Glasius et al.,
1999; Larsen et al., 2001; Yasmeen et al., 2012) are known to be present in α-pinene/O$_3$ SOA.
The proposed mechanism is consistent with the observation made by Zhang et al. (2015) that *cis*-
pinic acid is still generated after consumption of α-pinene upon ozonolysis, suggesting a particle-
phase production pathway. It is also in agreement with observations made by Lopez-Hilfiker et
al. (2015) and Huang et al. (2017), who examined the thermal behavior of α-pinene/O$_3$ SOA and
found that *cis*-pinic acid and hydroxypinonic acid can also arise from thermal decomposition of
unstable hetero-oligomers. In addition, it is consistent with the findings by Mutzel et al. (2015)
that intact HOMs detected in the gas phase are carbonyl-containing compounds. Recent work has
also established that hydroperoxides present in α-pinene/O$_3$ SOA are unstable and quickly
convert to more stable products (Krapf et al., 2016). Furthermore, monomers including *cis*-pinic
acid and terpenylic acid were found to be major constituents of the 10 and 20 nm particles from
α-pinene ozonolysis in a flow reactor (Winkler et al., 2012), which are likely fragments of high-
molecular-weight compounds due to the thermal decomposition of unstable hetero-oligomers
during resistive heating of particles in the thermal desorption chemical ionization MS
measurements (Hall and Johnston, 2012b).
**3.2.3. Formation mechanism proposed for the MW 358 ester**
A possible formation mechanism leading to the MW 358 ester is provided in Scheme 6.
Compared to the mechanism proposed for the MW 368 ester, an alkoxy radical related to 5-



hydroxypinonic acid instead of an alkoxy radical related to 7-hydroxypinonic acid participates in
the $RO_2\cdot + R'O\cdot \rightarrow RO_3R'$ radical termination reaction. The detailed mechanism leading to the
alkoxy radical related to 5-hydroxypinonic acid is given in Scheme S4 of the supplement. It is
noted that the $C_{19}H_{28}O_{11}$ dimeric HOM species is a positional isomer of that implicated in the
formation of the MW 368 ester  (Scheme 5). With regard to the suggestion that an isomeric
alkoxy radical is involved, a recent study by Zhang et al. (2017) provided evidence for the
formation of the corresponding hydroperoxy product, 5-hydroperoxypinonic acid, in α-pinene/$O_3$
SOA. As mentioned above, hydroxypinonic acids are major monomers in α-pinene/$O_3$ SOA (Fig.
S3), and it can be seen that at least two positional isomeric hydroxypinonic acids are present. To
arrive at the formation of the MW 358 ester [Fig. 1; structure (**a**)], a complex rearrangement
involving the labile inner part containing a linear trioxide function has to be invoked. A detailed
rearrangement mechanism is provided in Scheme S5 of the supplement. It can also be seen that
the labile intermediate (**c**) (Scheme 6) can serve as a precursor for *cis*-pinic acid, as it contains a
*cis*-pinic acid residue. In addition, it can be explained that this labile intermediate can also give
rise to the formation of terpenylic acid, a major monomer observed in α-pinene/$O_3$ SOA (Fig. S3)
(Claeys et al., 2009) but here again a complex rearrangement has to be invoked (Scheme S6 of
the supplement). In this context, there is evidence that unstable hetero-oligomers present in α-
pinene/$O_3$ SOA produce terpenylic acid upon heating (Lopez-Hilfiker et al., 2015). As already
mentioned above, terpenylic acid was also found to be a major constituent of the 10 and 20 nm
particles from α-pinene ozonolysis in a flow reactor (Winkler et al., 2012), which is likely formed
by decomposition of unstable hetero-oligomeric species in the thermal desorption chemical
ionization MS measurements (Hall and Johnston, 2012b).
[Scheme 6]

## 4.  Conclusions and atmospheric implications

Unambiguous mass spectrometric evidence has been obtained in this study for the linkage in the
MW 368 ($C_{19}H_{28}O_7$) hydroxypinonyl ester of *cis*-pinic acid, which is an abundant compound
present in α-pinene/$O_3$ SOA; more specifically, the MW 368 compound corresponds to an ester
of *cis*-pinic acid where the carboxyl substituent of the dimethylcyclobutane ring and not the
methylcarboxyl substituent is esterified with 7-hydroxypinonic acid. This linkage was already
proposed in previous work for the MW 358 ($C_{17}H_{26}O_8$) diaterpenylic ester of *cis*-pinic acid,



another major compound present in α-pinene/$O_3$ SOA (Yasmeen et al., 2010), but could be
supported in the present study. Guided by the molecular structures of these stable esters, we
propose a formation mechanism from highly oxygenated molecules in the gas phase that takes
into account the detection of an unstable $C_{19}H_{28}O_{11}$ HOM as a major species by CI-APi-TOF MS
with nitrate clustering (Ehn et al., 2012, 2014; Zhao et al., 2013; Krapf et al., 2016). It is
suggested that an acyl peroxy radical related to *cis*-pinic acid ($RO_2\cdot$) and an alkoxy radical related
to 7- or 5-hydroxypinonic acid ($R'O\cdot$) serve as key gas-phase radicals and combine according to a
$RO_2\cdot + R'O\cdot \rightarrow RO_3R'$ radical termination reaction. Subsequently, the unstable $C_{19}H_{28}O_{11}$
dimeric HOM species decompose by the loss of oxygen or ketene from the inner part containing
a labile linear trioxide bridge and the conversion of the unstable acyl hydroperoxide groups to
carboxyl groups, resulting in stable esters with a molecular composition of $C_{19}H_{28}O_7$ (MW 368)
and $C_{17}H_{26}O_8$ (MW 358), respectively. The proposed mechanism is supported by several
observations reported in the literature, one of them being that *cis*-pinic acid is still generated after
the consumption of α-pinene upon ozonolysis, suggesting its formation from an unstable HOM
species (Zhang et al., 2015).
Further theoretical investigations are warranted to examine the proposed mechanism leading to
the MW 368 and 358 esters present in α-pinene/$O_3$ SOA. The mechanism is assumed to be
energetically favorable as small stable molecules such as oxygen and ketene are expelled and a
stable ester bridge is generated. The mechanism involves the combination of an acyl peroxy with
an alkoxy radical according to a $RO_2\cdot + R'O\cdot \rightarrow RO_3R'$ reaction and thus differs from that
proposed to explain the formation of a MW 326 ester, where two acyl peroxy radicals related to
*cis*-pinic acid combine according to a $RO_2\cdot + RO_2\cdot \rightarrow ROOR + O_2$ reaction (Zhang et al., 2015).
We hypothesize that $RO_2\cdot + R'O\cdot \rightarrow RO_3R'$ chemistry is at the underlying molecular basis of
high-molecular-weight hetero-dimer formation in the gas phase upon α-pinene ozonolysis and
may thus be of importance for new particle formation and growth in pristine forested
environments.
**Acknowledgements**
Research at the University of Antwerp and TROPOS was supported by the Belgian Federal
Science Policy Office through the network project "Biogenic Influence on Oxidants and
Secondary Organic Aerosol: theoretical, laboratory and modelling investigations (BIOSOA)".



Research at the University of Antwerp was also supported by the Research Foundation – Flanders
(FWO), whereas research at TROPOS was also supported by the European Commission through
the EUROCHAMP-2 project (228335). We also would like to thank Anke Mutzel, Torsten
Berndt and Hartmut Herrmann from TROPOS for valuable discussions on the proposed
mechanism, and Olaf Böge for his help in the preparation of the α-pinene/$O_3$ SOA sample.

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




**Fig. 1.** Overview of the proposed high-molecular-weight ester compounds present in α-pinene/O$_3$
SOA which were investigated in the present study. The compounds present in underivatized α-
pinene/O$_3$ SOA are highlighted in red color.




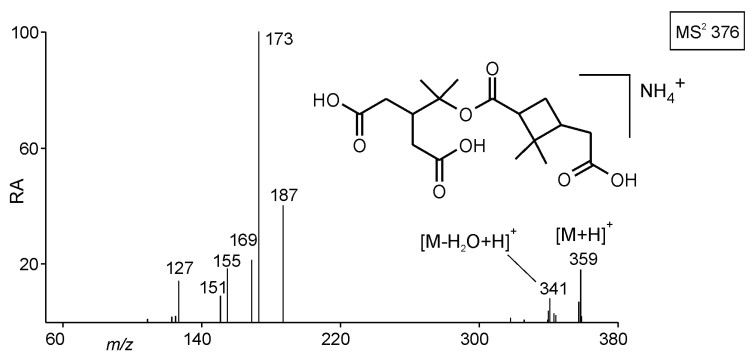

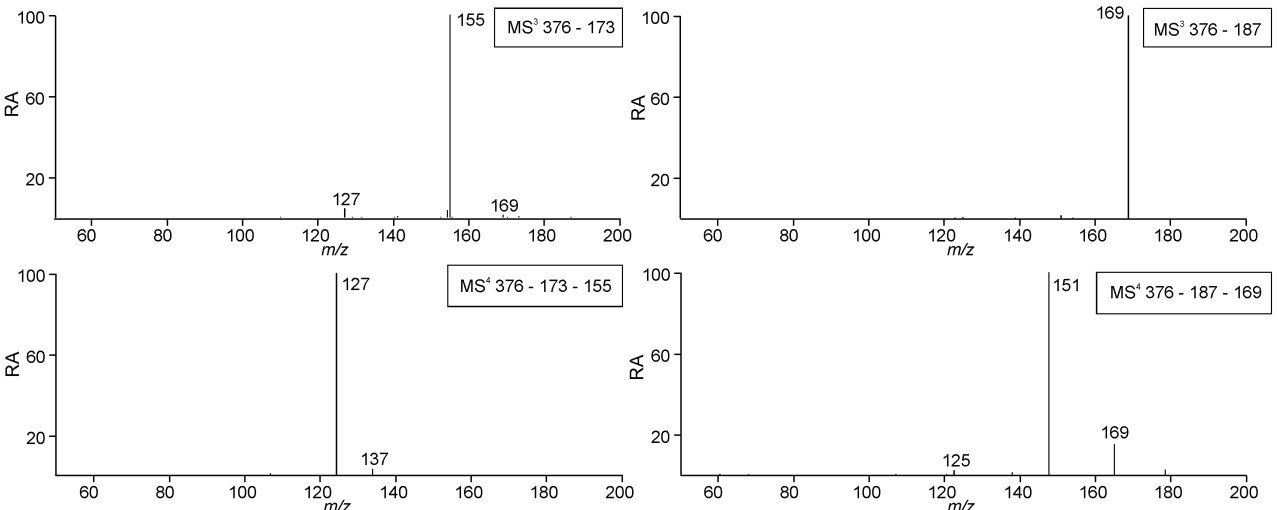


**Fig. 2.** Selected LC/(+)ESI-MS data for the non-derivatized dimeric MW 358 compound eluting at 24.7
min (Fig. S3) with its proposed structure in α-pinene/O$_3$ SOA, showing the MS$^2$ data for its ammonium
adduct ion at $m/z$ 376, $m/z$ 376 → $m/z$ 173 MS$^3$ data, $m/z$ 376 → $m/z$ 173 → $m/z$ 155 MS$^4$ data, $m/z$ 376
→ $m/z$ 187 MS$^3$ data, and $m/z$ 376 → $m/z$ 187 → $m/z$ 169 MS$^4$ data.





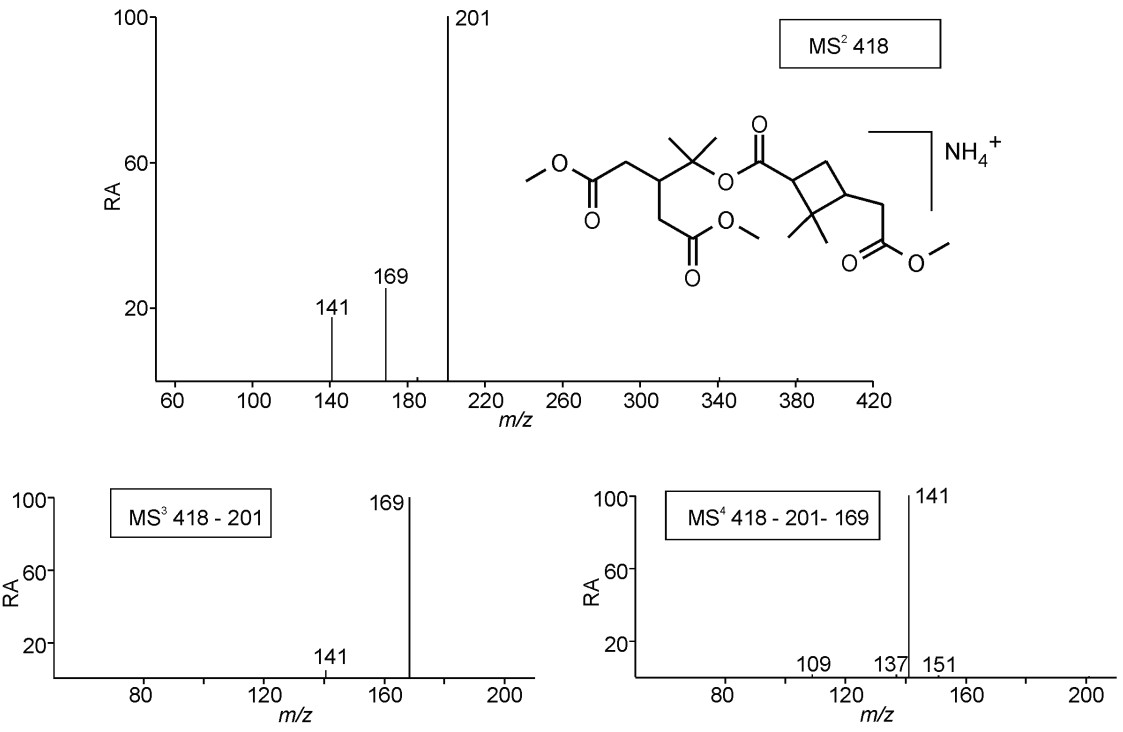


**Fig. 3.** Selected LC/(+)ESI-MS data for the trimethylated dimeric MW 358 compound eluting at 28.0
min (Fig. S4) with its proposed structure in α-pinene/O$_3$ SOA, showing  the MS$^2$ data for its ammonium
adduct ion at $m/z$ 418, $m/z$ 418 → $m/z$ 201 MS$^3$ data, and $m/z$ 418 → $m/z$ 201 → $m/z$ 169 MS$^4$ data.



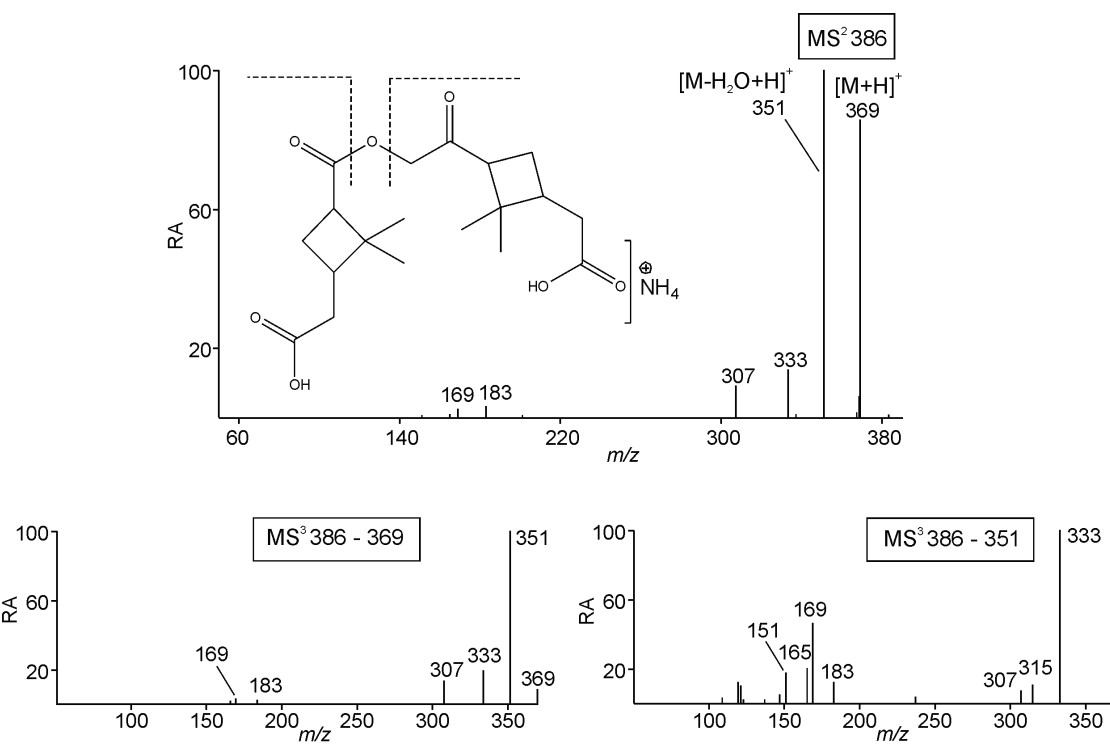


**Fig. 4.** Selected LC/(+)ESI-MS data for the non-derivatized dimeric MW 368 compound eluting at 25.9
min (Fig. S3) with its proposed structure in α-pinene/$O_3$ SOA, showing the $MS^2$ data for its ammonium
adduct ion at *m/z* 386, *m/z* 386 → *m/z* 369 $MS^3$ data and *m/z* 386 → *m/z* 351 $MS^3$ data.



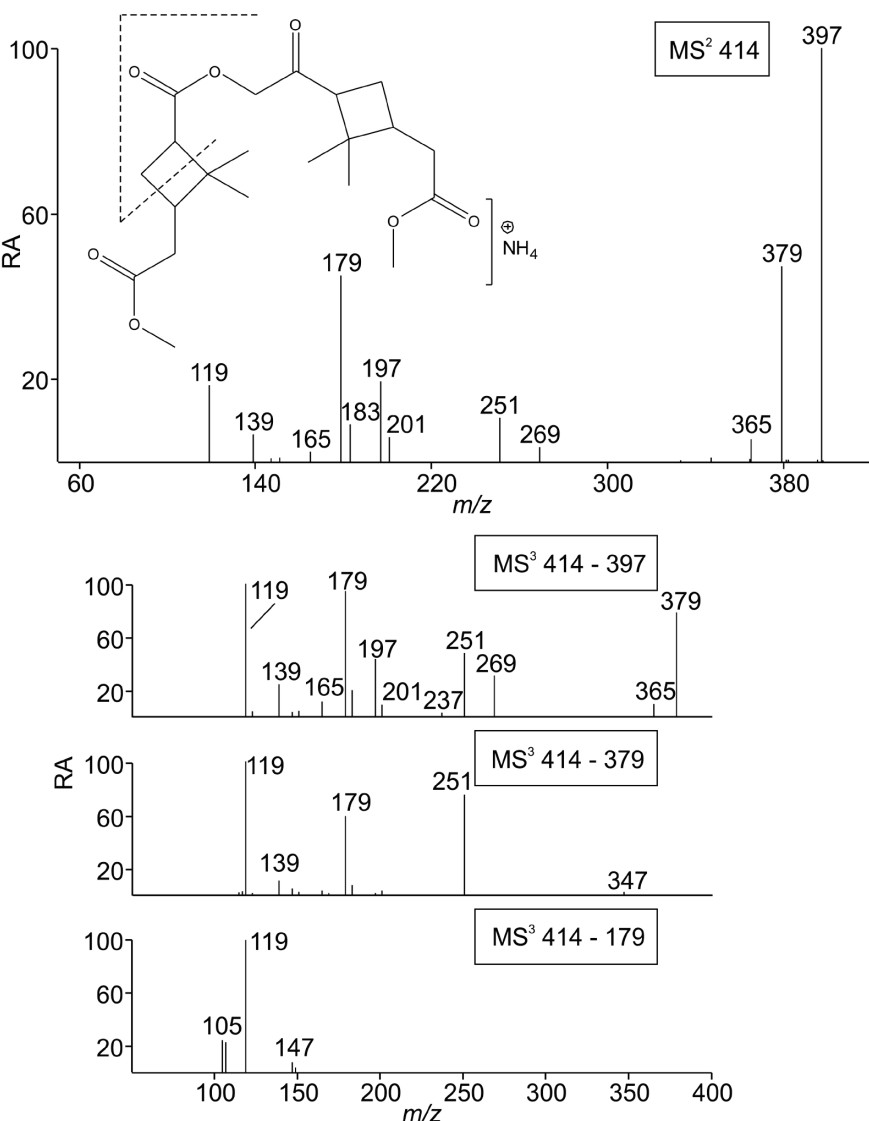


**Fig. 5.** Selected LC/(+)ESI-MS data for the dimethylated dimeric MW 368 compound eluting at 28.4
min (Fig. S4) with its proposed structure in α-pinene/O$_3$ SOA, showing the MS$^2$ data for its ammonium
adduct ion at *m/z* 414, *m/z* 414 → *m/z* 397 MS$^3$ data,  *m/z* 414 → *m/z* 379 MS$^3$ data, and *m/z* 414 → *m/z*
179 MS$^3$ data.


**Scheme 1.** Proposed fragmentation mechanism for the ammoniated non-derivatized MW 358 ester present in α-pinene/O₃ SOA.

**Scheme 2.** Proposed fragmentation mechanism for the ammoniated MW 358 ester trimethylated derivative.





**Scheme 3.** Proposed fragmentation mechanism for the ammoniated non-derivatized MW 368 ester present in α-pinene/O₃ SOA.






**Scheme 4.** Proposed fragmentation mechanism for the ammoniated MW 368 ester dimethylated derivative.

651
652

653

654





**Scheme 5.** Proposed simplified mechanism leading to the formation of the MW 368 ester with structure (**b**) (Fig. 1). The mechanisms suggested for the formation of the alkoxy radical related to 7-hydroxypinonic acid (**a**) and the acyl peroxy radical related to *cis*-pinic acid (**b**) are provided in Scheme S4 of the supplement. It is proposed that the latter radicals serve as key intermediates. Radical termination according to a RO$_2$·+ R'O·→ RO$_3$R' reaction results in a HOM with a molecular composition of C$_{19}$H$_{28}$O$_{11}$ (**c**), a major gas-phase species upon α-pinene ozonolysis which has been detected by CI-APi-TOF MS (Ehn et al., 2012, 2014; Krapf et al., 2016). Further degradation of the labile inner part containing a linear trioxide bridge through the loss of oxygen and conversion of the acyl hydroperoxide groups to carboxyl groups results in the MW 368 ester.











**Scheme 6.** Proposed simplified mechanism leading to the formation of the MW 358 ester with structure (**a**) (Fig. 1). The mechanisms
suggested for the formation of the acyl peroxy radical related to *cis*-pinic acid (**a**) and the alkoxy radical related to 5-hydroxypinonic acid (**b**)
are provided in Scheme S4 of the supplement. It is proposed that the latter radicals serve as key intermediates. Radical termination according
to a $RO_2\cdot + R'O\cdot \rightarrow RO_3R'$ reaction results in a HOM with a molecular composition of $C_{19}H_{28}O_{11}$ (**c**), a major gas-phase species upon α-pinene
ozonolysis which has been detected by CI-APi-TOF MS (Ehn et al., 2012, 2014; Krapf et al., 2016). Further degradation of the labile inner
part containing a linear trioxide bridge through the loss of ketene and conversion of the acyl hydroperoxide groups to carboxyl groups results
in the MW 358 ester.