# Peer review of "Structural characterization and mechanistic proposal for their formation from highly"

_Atmospheric Chemistry and Physics, 2017_

## Referee Comment (RC1) · Anonymous Referee #1 · 9 Mar 2018

First, I would like to say to the authors and to anyone else who reads interactive comments how impressed I am with the experimental design and interpretation of MSn data. A lot of hard work went into this study. Thanks for an enjoyable read!

Interpretation of MSn data in general, and product ion signal intensities in particular, is impeded by the fact that ions readily undergo structural rearrangements that often can't be predicted ahead of time, causing misinterpretation. The strength of this study is the

presentation of two different sets of data (with/without methylation) for two different species (MW 358 and 368), all of which point to the same conclusion with regard to positional isomers. This gives substantial confidence in the conclusions.

Specific comments and questions:

Please comment about why positive ion ammonia adducts were studied as opposed to protonated or metal cationized molecules (positive spectra) or deprotonated molecules (negative spectra). I'm guessing that the additional fragmentation step of the ammonia adducts (initial loss of ammonia) takes away some internal energy of the ions, making subsequent fragmentation of MH+ more controlled and easier to interpret. Also, how intense are the ammonia adducts of these ions relative to other adducts? Was the solvent composition modified to enhance ammonia adduction?

Scheme 1. Should the dotted line for pathway b in the m/z 359 structure be pointed to the left rather than to the right?

Page 11, line 296. While I appreciate the authors point regarding the strength of the signal intensity of m/z 351 with respect to the two positional isomers, I think a much more compelling observation is the high intensity of m/z 333 ion (loss of the second H2O molecule), which would be much harder for the MW 368 structure a in Figure 1.

Page 12, lines 314-319. This discussion (fragmentation of the cyclobutane ring) is perhaps the weakest of the arguments made by the authors regarding positional isomers since it is based on previous measurements of the fragmentation of monomer compounds in negative ion mode. Nonetheless, it is consistent with other fragmentation products for MW 368 that point to positional isomer b in Figure 1.

Figures 4 and 5. I found the dotted lines within the structures in these figures to be distracting. The lines do indicate where the molecule is ultimately going to break, but the ammonia adduct decomposes to the protonated molecule first, which is not clear from the structure. I think it is better to keep the format of these structures similar to

Figures 2 and 3.

Page 12, section 3.2.1. Before launching into the gas phase formation mechanism, the authors should state (if it is the case) that there are no direct analogues to MW 358, 368 in the CI-APiTOF mass spectra. If MW 358 and 368 were produced in the gas phase, do you expect they would be sensitively detected by CI-APiTOF?

Sections 3.3.2 and 3.3.3. My understanding of gas phase radical-radical chemistry is that these types of reactions are relatively unconstrained - they occur with high probability and it is mostly a matter of which two specific species happen to collide first. With that in mind, it seems that this discussion and accompanying schemes 5 and 6 give a rationalization for the formation of MW 358 and 368, but they do not explain the preference for the specific positional isomers that were identified.

Given that identification of positional isomers is a key element of this study, can the authors explain this preference? Furthermore, why do you have the expectation that MW 358 and 368 must be produced from dimer precursors that were initially formed in the gas phase? Could it be possible that these species are formed from reactive nonvolatile monomers (peroxy compounds?) that condense to the particle phase and then react very quickly to make the dimer products? It seems that the positional isomer preference demonstrated by the MSn work might be exploited more fully to perhaps rule in/out certain reaction pathways.

---

## Referee Comment (RC2) · Anonymous Referee #2 · 5 Apr 2018

The manuscript describes the interpretation of LC-MS and MS-MS investigations from reaction chamber experiments of alpha-pinene ozone experiments. The focus is the structural characterization of higher molecular weight dimer esters. One of the conclusions of the manuscript is the proposal of a connection between the formation of the dimeric esters and the formation of highly oxidized dimers often observed in CIMS measurements. In general, the paper is well written and presents an interesting topic that is well suited to be published in ACP. The formation of higher molecular weight

compounds is still a topic of considerable interest, especially in connection with atmospheric new particle formation. Although the first observation of the dimers in biogenic SOA is already 20 years ago, the detailed structures and especially the formation pathways of these compounds are still unclear. In my opinion the most interesting aspect of the manuscript is the suggestion of an reaction of a peroxy- and an alkoxy-radical to form a dimer with a trioxide bridge, followed by a decomposition to form the observed stabile ester dimers. Such a pathway would finally bridge the observation of gas phase HOMs with the well-known dimer esters as identified by the traditional trace analytical community (e.g. using LC-MS). Therefore, in my opinion the manuscript is well suited to be published in ACP after considering the following minor comments.

Minor comments: The suggestion that an peroxy radical is involved in the formation of the dimers also fits to the observation of an suppression of NPF as observed in: Wildt, J., Mentel, T. F., Kiendler-Scharr, A., Hoffmann, T., Andres, S., Ehn, M., Kleist, E., Müsgen, P., Rohrer, F., Rudich, Y., Springer, M., Tillmann, R., and Wahner, A.: Suppression of new particle formation from monoterpene oxidation by NOx, Atmos. Chem. Phys., 14, 2789-2804, https://doi.org/10.5194/acp-14-2789-2014, 2014. The authors might consider to also refer to this work.

I might have missed that in the text but are there indications if the final dimeric ester products (i.e. MW 358 and 368) can also be detected in the gas phase? If not, do the authors believe that this is a consequence of the (non) ionization of the esters in their measurements or an indication that the formation of the final products takes place in the particle phase?

---

## Author Response (AR1)

**Responses to the comments of reviewer #1 and manuscript changes**

We would like to thank this reviewer for thoughtful and constructive comments, and for the general appreciation of the study.

Our responses to the specific comments and manuscript changes are as follows:

(1) Reviewer: Please comment about why positive ion ammonia adducts were studied as opposed to protonated or metal cationized molecules (positive spectra) or deprotonated molecules (negative spectra). I'm guessing that the additional fragmentation step of the ammonia adducts (initial loss of ammonia) takes away some internal energy of the ions, making subsequent fragmentation of MH+ more controlled and easier to interpret. Also, how intense are the ammonia adducts of these ions relative to other adducts? Was the solvent composition modified to enhance ammonia adduction?

(2) Response: The simple reason why in the positive ion mode ammonia adducts were selected is that under the LC conditions used these ions were more abundant than the protonated molecules, while metal-cationized molecules were not observed. The ion abundance ratios $[M + NH_4]^+/[M + H]^+$ were 13.0, 15.6, 38 and 7.7, for the MW 358 ester, MW 368 ester, MW 358 ester trimethylated derivative and MW 368 ester dimethylated derivative, respectively. Ammonia adducts were formed because the buffered aqueous LC mobile phase with pH 3 contained ammonium formate (50 mM). The LC solvent composition was thus not modified to enhance ammonia adduction. Upon collisional activation in the ion trap, the ammoniated adducts lose ammonia, resulting in the protonated molecules, of which the fragmentation can readily be explained.

(3) Manuscript changes: The following sentences have been added to Section 2.3 of the experimental part (lines 178 – 184): "Under the LC conditions used, ammoniated adducts were detected owing to the presence of ammonium formate in mobile phase (A). The ion abundance ratios $[M + NH_4]^+/[M + H]^+$ were 13.0, 15.6, 38 and 2.3, for the MW 358 ester, MW 368 ester, MW 358 ester trimethylated derivative and MW 368 ester dimethylated derivative, respectively. In the ion trap $MS^n$ experiments, ammonia adducts were selected as precursor ions because these ions were more abundant than the protonated molecules and lose ammonia upon $MS^2$, resulting in protonated molecules of which the fragmentation can be readily explained."

(1) Reviewer: Scheme 1. Should the dotted line for pathway b in the m/z 359 structure be pointed to the left rather than to the right?

(2) Response: Yes, this is indeed wrong. Scheme 1 will be corrected in the revised manuscript.

(3) Manuscript changes: A corrected scheme 1 has been included.

(1) reviewer: Page 11, line 296. While I appreciate the authors point regarding the strength of the signal intensity of m/z 351 with respect to the two positional isomers, I think a much more compelling observation is the high intensity of m/z 333 ion (loss of the second H2O molecule), which would be much harder for the MW 368 structure a in Figure 1.

(2) Response: We thank the reviewer for this input. The main text will be changed accordingly to include this observation. The loss of a second molecule of water would indeed be more difficult to explain with positional isomer a.

(3) Manuscript changes: The text has been modified/extended as follows (lines 267 – 272): "It is noted that most ions discussed above can also be explained with a positional isomeric structure [Fig. 1; structure (**a**)], although we would expect that such a structure would lead to a less pronounced loss of water from $m/z$ 369 resulting in $m/z$ 351, as it would result in strain in the dimethylcyclobutane ring. In addition, the formation of $m/z$ 333, involving a second loss of water, supports the proposed isomeric structure (**b**) (Fig. 1), as this process is more difficult to explain with isomeric structure (**a**)."

(1) Page 12, lines 314-319. This discussion (fragmentation of the cyclobutane ring) is perhaps the weakest of the arguments made by the authors regarding positional isomers since it is based on previous measurements of the fragmentation of monomer compounds in negative ion mode. Nonetheless, it is consistent with other fragmentation products for MW 368 that point to positional isomer b in Figure 1.

(2) Response: The discussion of the fragmentation of the dimethylcyclobutane ring is indeed based on previous measurements of the fragmentation of monomer compounds (pinic acid and norpinic acid) in negative ion mode but we would like to retain this argument in favor of positional isomer b.

(3) Manuscript changes: no changes were made.

(1) Reviewer: Figures 4 and 5. I found the dotted lines within the structures in these figures to be distracting. The lines do indicate where the molecule is ultimately going to break, but the ammonia adduct decomposes to the protonated molecule first, which is not clear from the structure. I think it is better to keep the format of these structures similar to Figures 2 and 3.

(2) Response: For clarity, the dotted lines will be removed in Figures 4 and 5. We fully agree with the reviewer.

(3) Manuscript changes: The dotted lines have been removed in Figures 4 and 5.

(1) Reviewer: Page 12, section 3.2.1. Before launching into the gas phase formation mechanism, the authors should state (if it is the case) that there are no direct analogues to MW 358, 368 in the CI-APiTOF mass spectra. If MW 358 and 368 were produced in the gas phase, do you expect they would be sensitively detected by CI-APiTOF?

(2) Response:  As suggested, it will be stated that there are no direct analogues to the MW 358 and 368 esters in the CI-APiTOF mass spectra. But it is hard to predict whether the CI-APiTOF MS technique with nitrate clustering would be sensitive to the detection of these compounds. As far as we understand the CI-APiTOF MS technique is very sensitive to the detection of peroxy compounds.

(3) Manuscript changes: The following sentence has been added (lines 306 - 308): "It is noted that the CI-APi-TOF MS technique does not reveal $C_{19}$ HOM species that correspond to direct analogues of the MW 358 and 368 esters."

(1) Sections 3.3.2 and 3.3.3. My understanding of gas phase radical-radical chemistry is that these types of reactions are relatively unconstrained - they occur with high probability and it is mostly a matter of which two specific species happen to collide first. With that in mind, it seems that this discussion and accompanying schemes 5 and 6 give a rationalization for the formation of MW 358 and 368, but they do not explain the preference for the specific positional isomers that were identified.

(2) Response: We believe that the discussion accompanying schemes 5 and 6 provides a reasonable rationalization for the formation of the MW 358 (isomer a) and MW 368 (isomer b). For the formation of the other positional isomers , one would have to consider the formation of isomeric monomeric radicals. For example, the formation of MW 368 (isomer a) would require the formation of a C10H15O5 radical, containing the carboxyperoxy radical at the opposite methylcarboxyperoxy substituent and would require one or more steps.

(3) Manuscript changes: No changes were made.

(1) Reviewer: Given that identification of positional isomers is a key element of this study, can the authors explain this preference? Furthermore, why do you have the expectation that MW 358 and 368 must be produced from dimer precursors that were initially formed in the gas phase? Could it be possible that these species are formed from reactive nonvolatile monomers (peroxy compounds?) that condense to the particle phase and then react very quickly to make the dimer products? It seems that the positional isomer preference demonstrated by the MSn work might be exploited more fully to perhaps rule in/out certain reaction pathways.

(2) Response: As discussed in the manuscript, we propose that the MW 358 and 368 esters are produced from unstable dimer precursors that are initially formed in the gas phase. An argument in favor of this hypothesis, mentioned in the manuscript,  is that the proposed gas phase precursors, i.e., isomeric C19 peroxy species, are unstable (lines 327 -  332). This is supported by an alpha-pinene ozonolysis flow tube experiment reported by Krapf et al. (2016), where in addition to the C19H28O11 species also C19 species are detected corresponding to the conversion of one acyl hydroperoxy group (C19H28O10), the conversion of two acyl hydroperoxy groups (C19H28O9), the loss of oxygen (C19H28O9), and the loss of oxygen combined with the conversion of one acyl hydroperoxy group (C19H28O8). As clearly stated in the manuscript, the proposed formation pathways should be regarded as tentative.

We think that it is rather unlikely that the MW 358 and 368 esters are formed in the condensed phase by reactive monomers containing peroxy groups. In an earlier study (Yasmeen et al., 2010) we have proposed that the MW 358 ester is produced in the condensed phase by reaction between pinic acid and the lactone-containing terpenylic acid, but a subsequent study by Kristensen et al. (2014) provided evidence that this pathway is not supported and that a gas-phase formation pathway should be considered. In the cited study, the formation of dimers has been shown to occur only during ozonolysis of α-pinene and not through OH-oxidation, suggesting that these dimers originate via the gas-phase reaction of a stabilized Criegee intermediate formed by ozonolysis of α-pinene. This mechanism is supported by the increase in the formation of dimers observed at higher RH, explained by increased stabilization of the Criegee intermediate. This has been addressed in the introduction of the original version of the manuscript (lines 130-136), where we write: "Efforts to understand ester formation from α-pinene ozonolysis have also been actively undertaken. Yasmeen et al. (2010) proposed that ester formation took place in the particle phase by esterification of cis-pinic acid with terpenylic acid but this mechanism was not retained in later studies. Kristensen et al. (2014) demonstrated their formation through gas-phase ozonolysis and supported the participation of a stabilized Criegee intermediate, as previously suggested for the formation of unstable high-molecular-weight compounds that play a role in new particle formation (Ziemann, 2002; Bonn et al., 2002; Lee and Kamens, 2005)."

(3) Manuscript changes: No changes were made in the manuscript. In our opinion, we have provided sufficient arguments for the formation of the MW 358 and 368 through a gas-phase mechanism.

**Responses to the comments of reviewer #2 and manuscript changes**

We would like to thank the reviewer for thoughtful and constructive comments, and for the general appreciation of the study.

Our responses to the specific comments are as follows:

(1) Reviewer: The suggestion that an peroxy radical is involved in the formation of the dimers also fits to the observation of an suppression of NPF as observed in: Wildt, J., Mentel, T. F., Kiendler-Scharr, A., Hoffmann, T., Andres, S., Ehn, M., Kleist, E., Müsgen, P., Rohrer, F., Rudich, Y., Springer, M., Tillmann, R., and Wahner, A.: Suppression of new particle formation from monoterpene oxidation by NOx, Atmos. Chem. Phys., 14, 2789-2804, https://doi.org/10.5194/acp-14-2789-2014, 2014. The authors might consider to also refer to this work.

(2) Response: We thank the reviewer for this input. This study will be mentioned in the revised manuscript.

(3) Manuscript changes: The following sentence has been added in the introduction (lines 132-134): "In addition, the suggestion that peroxy radicals are involved in the formation of dimers also fits to the observation of a suppression of new particle formation from monoterpene oxidation by $NO_x$ (Wildt et al., 2014)." The reference has been included in the reference list.

(1) Reviewer: I might have missed that in the text but are there indications if the final dimeric ester products (i.e. MW 358 and 368) can also be detected in the gas phase? If not, do the authors believe that this is a consequence of the (non) ionization of the esters in their measurements or an indication that the formation of the final products takes place in the particle phase?

(2) Response: We are not aware that the final dimeric MW 368 and 358 esters can also be detected in the gas phase by the CI-API-TOF technique with nitrate clustering. It is well possible that the latter technique is not sensitive to the detection of the MW 368 and 358 esters. As far as we understand the CI-APiTOF MS technique is very sensitive to the detection of peroxy compounds. A similar comment was also made by reviewer #1. In response, as suggested by this reviewer, it will be stated in the revised version that there are no direct analogues to the MW 358 and 368 esters in the CI-APiTOF mass spectra. It is hard to say at which stage (gas or condensed phase) and to which extent the final products will be formed from the unstable gas-phase peroxy precursors but there is evidence from an α-pinene ozonolysis flow tube experiment reported by Krapf et al. (2016) that degradation already takes place under the flow tube conditions: in addition to the $C_{19}H_{28}O_{11}$ species also C19 species are detected corresponding to the conversion of one acyl hydroperoxy group ($C_{19}H_{28}O_{10}$), the conversion of two acyl hydroperoxy groups ($C_{19}H_{28}O_{9}$), the loss of oxygen ($C_{19}H_{28}O_{9}$), and the loss of oxygen combined with the conversion of one acyl hydroperoxy group ($C_{19}H_{28}O_{8}$).

(3) Manuscript changes: It has now been more clearly stated in the revised manuscript that there are no direct analogues to the MW 358 and 368 esters in the CI-APiTOF mass spectra. See our response to the comment made by reviewer #1.

**No changes were made in the supplement.**

[revised manuscript text omitted]